# Since Faithfulness Fails: The Performance Limits of Neural Causal Discovery

## Abstract

Neural causal discovery methods have recently improved in terms of scalability and computational efficiency. However, there are still opportunities for improving their accuracy in uncovering causal structures. We argue that the key obstacle in unlocking this potential is the *faithfulness assumption*, commonly used by contemporary neural approaches. We show that this assumption, which is often not satisfied in real-world or synthetic datasets, limits the effectiveness of existing methods. We evaluate the impact of faithfulness violations both qualitatively and quantitatively and provide a unified evaluation framework to facilitate further research.

## 1 Introduction

Causal discovery is essential to scientific research, driving a growing demand for machine learning methods to support this process. Despite the development of several neural-based causal discovery methods in recent years (Brouillard et al., 2020; Lorch et al., 2021; Annadani et al., 2023; Nazaret et al., 2024), their performance remains insufficient for real-world applications, particularly in fields like medicine and biology (de Castro et al., 2019; Peters et al., 2016). Furthermore, these methods are usually evaluated using synthetic datasets, which vary between studies, obscuring the overall picture and making assessment of advancements difficult.

To address this challenge, we introduce a unified benchmark for evaluating neural causal discovery methods. Specifically, we use identical datasets, tune hyperparameters consistently, and use a standardized functional approximation across all methods. Our systematic evaluation reveals that, although there has been progress in computational efficiency over the past few years, significant gains in causal discovery accuracy have yet to emerge. Further underscoring the challenges, we discover that the existing methods can not take advantage of the increasing amount of data, countering the universally held assumption that more data leads to better learning.

The key claim of this work is that progress in causal discovery requires moving beyond the faithfulness assumption. Although it is widely known that real-world and synthetic data rarely satisfy this assumption (Hoover, 2001; Andersen, 2013), most neural-based methods overlook its impact. We develop techniques to measure how faithfulness violations degrade performance and set an upper bound for current benchmarks. Our results show a clear correlation: faithfulness violations significantly hinder performance, and improvements within the current paradigm are limited.

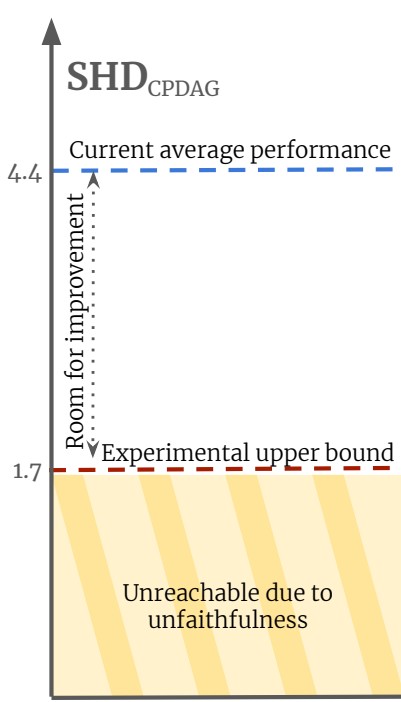

Figure 1: Neural causal discovery methods suffer from inherent performance limit due to violation of faithfulness assumption, but there is still room for improvement. Values computed for ER(5,1) class of graphs. See Sections 3, 5.

We believe that our work establishes a solid foundation that will propel future research in ML methods for causal discovery. Our original contributions are as follows:

- We identify violations of faithfulness as the core challenge and analyze its consequences both qualitatively and quantitatively.
- We develop an open unified benchmark for causal discovery evaluation.
- We present a soft upper bound on the performance of neural causal discovery methods for synthetic benchmarks.

## 2 BACKGROUND

**Structural Causal Models (SCMs) and graph representation**    Causal relationships are commonly formalized using SCMs, which represent causal dependencies through a set of structural equations. For a directed acyclic graph (DAG) $G = (V, E)$, an SCM is defined by a set of equations

$$X_i = f_i(Pa_i, U_i), \tag{1}$$

where $i \in V$, $X_i$ is a random variable, $f_i \colon \mathbb{R}^{|Pa_i|+1} \to \mathbb{R}$ is a function, $Pa_i$ denotes the set of parents of the vertex $i$ in the graph $G$, and $U_i$ is an independent noise term associated with $X_i$. In this paper, we assume *additive noise* SCMs, also referred to as *additive noise models* (ANM), where:

$$f_i(Pa_i, U_i) = g_i(Pa_i) + U_i \tag{2}$$

for some $g_i \colon \mathbb{R}^{|Pa_i|} \to \mathbb{R}$.

**Causal discovery**    Causal structure discovery aims to recover the ground truth DAG representing causal relationships among variables. However, the unique solution cannot be indentified from the observational data only; instead, one can only identify the structure up to a Markov Equivalence Class (MEC), the set of DAGs that encode the same conditional independencies. This can be uniquely represented by a Complete Partially Directed Acyclic Graph (CPDAG), which is a sum of DAGs from the same class. This results in a graph that includes both directed and undirected edges, reflecting consistent and uncertain causal directions within the MEC (Verma & Pearl, 1990).

**Faithfulness assumption**    A probability distribution $P$ is said to be *faithful* to a DAG $G = (V, E)$ if all the conditional independence relations present in the data correspond to those implied by the $d$-separation criteria of the DAG (for more on $d$-separation, see Appendix A.1 or Pearl (2009)). Formally, this can be written as:

$$X_a \perp\!\!\!\perp X_b \mid X_S \quad \Rightarrow \quad a \text{ is } d\text{-separated from } b \mid S, \tag{3}$$

where $\perp\!\!\!\perp$ denotes conditional independence of the variables, $a, b \in V$ are nodes of the graph, and $S \subseteq V \setminus \{a, b\}$ is a set of nodes. Intuitively, the faithfulness assumption can be understood as the statement that all statistical independencies in the observed data are the result of the underlying causal structure. Faithfulness assumption can be violated, for example, in a situation when paths cancel each other effects out, leading to statistical independence despite an existing causal relationship. An example of this kind of violation is shown in Appendix A.2.

While the faithfulness is a useful and powerful assumption in causal discovery, it is rarely satisfied in the practical scenarios (Cartwright, 2001; Andersen, 2013).

**Score-based neural causal discovery**    To allow for scalable causal discovery on graphs with hundreds of nodes, recent approaches focus on heuristics employing continuous optimization techniques that use neural networks as functional approximators to model the underlying probability distribution of the data (Nazaret et al., 2024). These approaches use a continuous representation of the graph structure, enforcing a differentiable acyclicity constraint to ensure the result is a valid DAG. The primary objective is to maximize $\log p_\theta(X|G)$, that is the log-likelihood of the data given the graph while incorporating regularization terms to control graph complexity. The training procedure comprises two parts: fitting functional approximators and structure search. They are usually done in parallel to maximize compute efficiency. Methods of this class are guaranteed to recover a DAG from the MEC class of ground true graph when the faithfulness assumption is fulfilled (see Brouillard et al. (2020)).

We benchmark four differentiable causal discovery methods DCDI (Brouillard et al., 2020), SDCD (Nazaret et al., 2024), BayesDAG (Annadani et al., 2023), and DiBS (Lorch et al., 2021), as they summarize various research directions and improvements explored in neural causal discovery over the last four years (see Appendix E). DCDI and SDCD represent the graph using an adjacency matrix, and optimize using the Augmented Lagrangian method (Zheng et al., 2018), aiming to find a single graph that maximizes the likelihood, with regularization added to penalize complex structures. In contrast, BayesDAG and DiBS take a Bayesian approach, approximating the posterior distribution over graphs rather than finding a single solution, with regularization introduced via prior distributions on graph structures. All four methods assume that the distribution is faithful to the ground truth DAG.

**Structure evaluation**  We evaluate graph discovery within the MEC using $\text{ESHD}_{\text{CPDAG}}$ and $\text{F1-Score}_{\text{CPDAG}}$, where $\text{ESHD}_{\text{CPDAG}} = 0$ and $\text{F1-Score}_{\text{CPDAG}} = 1$ when the predicted graph is in the same MEC as the ground truth. For Bayesian methods, we compute the average by sampling 100 graphs from the posterior; for non-Bayesian methods, we use a single graph.

The Structural Hamming Distance (SHD) (Tsamardinos et al., 2006) counts edge insertions, deletions, and reversals needed to match the predicted graph to the true graph. We define **Expected SHD between CPDAGs** as:

$$\text{ESHD}_{\text{CPDAG}}(\mathcal{G}, \mathbb{G}) = \mathbb{E}_{\mathcal{G}^* \sim \mathbb{G}}[\text{SHD}(\text{CPDAG}(\mathcal{G}), \text{CPDAG}(\mathcal{G}^*))], \tag{4}$$

where $\mathbb{G}$ is the resulting distribution of graphs, $\mathcal{G}^*$ is a graph sampled from $\mathbb{G}$ and $\mathcal{G}$ is the ground true graph. The F1-Score measures the harmonic mean of precision and recall for edge predictions. We compute the **Expected F1-Score between the CPDAGs** as follows:

$$\text{F1-Score}_{\text{CPDAG}}(\mathcal{G}, \mathbb{G}) = \mathbb{E}_{\mathcal{G}^* \sim \mathbb{G}}[\text{F1-Score}(\text{CPDAG}(\mathcal{G}), \text{CPDAG}(\mathcal{G}^*))]. \tag{5}$$

For more details and justification on the selection of metrics please refer to Appendix D.

# 3 UNIFIED BENCHMARK FOR SCORE-BASED NEURAL CAUSAL DISCOVERY METHODS ON SYNTHETIC DATA

In this section, we present a unified benchmark that exposes both the strengths and limitations of neural-based causal discovery methods. We evaluate methods DiBS, DCDI, BayesDAG, and SDCD introduced in Section 2 on identical datasets, tune hyperparameters consistently, and use a common functional approximation.

Our analysis spans several key dimensions of performance. In Section 3.2, we show that despite advancements in causal discovery over the past few years, $\text{ESHD}_{\text{CPDAG}}$ and $\text{F1-Score}_{\text{CPDAG}}$ metrics do not improve significantly. In Section 3.3, we demonstrate that structure discovery accuracy does not scale with the amount of data. Finally, in Section 3.4, we confirm that variations in MLP architecture have minimal impact on performance. In Appendix F we provide additional results on real-world structures which align with the conclusions presented in this section.

## 3.1 EXPERIMENTAL SETUP

**Dataset generation**  We sample three types of graphs from the Erdős-Rényi (ER) distribution (Erdös & Rényi, 1959): one with 5 nodes and the expected degree of 1, another with 10 nodes and the expected degree of 2, and the third with 30 nodes and the expected degree of 2. These datasets are referred to as ER(5, 1), ER(10, 2), and ER(30, 2), respectively. These parameter choices align with commonly studied medium-sized graphs in causal discovery research (Brouillard et al., 2020; Nazaret et al., 2024). Data generation follows the SCM formalism introduced in Section 2, with functional relationships modeled by two-layer neural networks (hidden dimension 8, ReLU activation) and additive Gaussian noise. The noise has zero mean, and its variance is sampled independently for each node. This setup is known to be challenging (Geffner et al., 2024; Nazaret et al., 2024). For more details refer to Appendix C.1.

| Method | ER(5, 1) | | ER(10, 2) | | ER(30, 2) | |
|---|---|---|---|---|---|---|
| | ESHD$_{CPDAG}$ | F1-Score$_{CPDAG}$ | ESHD$_{CPDAG}$ | F1-Score$_{CPDAG}$ | ESHD$_{CPDAG}$ | F1-Score$_{CPDAG}$ |
| DCDI | 5.7 (3.7, 8.1) | 0.60 (0.46, 0.74) | **16.9** (15.7, 18.1) | 0.52 (0.50, 0.56) | **45.9** (42.0, 49.9) | **0.73** (0.69, 0.77) |
| BayesDAG | 3.9 (3.6, 4.3) | 0.78 (0.77, 0.81) | 18.3 (16.9, 19.8) | **0.56** (0.54, 0.59) | 51.7 (48.2, 55.9) | 0.59 (0.57, 0.61) |
| DiBS | **2.6** (1.7, 3.7) | **0.85** (0.80, 0.90) | 16.9 (14.2, 201) | 0.61 (0.57, 0.68) | 68.0 (65.3, 70.9) | 0.23 (0.22, 0.24) |
| SDCD | 5.4 (3.8, 6.7) | 0.60 (0.35, 0.69) | 20.9 (19.5, 22.2) | 0.54 (0.46, .62) | 62.8 (58.8, 67.7) | 0.55 (0.53, 0.58) |

Table 1: Comparison of ESHD$_{CPDAG}$ and F1-Score$_{CPDAG}$ for different methods on ER(10, 2) (left) and ER(30, 2) (right) dataset. The numbers in the subscripts correspond to 95% confidence intervals. The statistics were computed based on 30 graphs.

**Hyperparameter tuning**   To ensure a fair comparison across all methods, we perform systematic hyperparameter tuning, selecting the best-performing parameters for each model. We employ a grid search approach based on the parameter ranges suggested by the original authors. This process optimizes key variables such as regularization coefficients, sparsity controls, and kernel configurations. Details can be found in Appendix C.2.

**Functional approximators**   We standardize the choice of functional approximators across all experiments, using a two-layer MLP with a hidden dimension of 4. This model size is consistent with previous work (Brouillard et al., 2020; Nazaret et al., 2024) and has proven to perform well across all the benchmarked methods, as discussed in Section 3.4. Additionally, we use trainable variance to allow the model to adapt to varying noise levels, in line with our dataset generation setup.

## 3.2   PERFORMANCE COMPARISON

Table 1, summarizes the benchmark results of neural-based causal discovery methods on graphs from ER(5, 1), ER(10, 2), and ER(30, 2) classes. We tune hyperparameters to optimize the ESHD$_{CPDAG}$ metric. For all classes of graphs, metrics were computed based on 30 graphs.

The results show that DiBS is particularly effective for smaller graphs (ER(5, 1) and ER(10, 2)), while DCDI is able to achieve the best results for moderate-size graphs (ER(10, 2) and ER(30, 2)). The ranking of the methods changes with the size of the graphs but SDCD consistently exhibits the worst performance in terms of ESHD$_{CPDAG}$. Nevertheless, the performance of all the methods remains unsatisfactory with all methods predicting more than half of the edges incorrectly.

## 3.3   IMPACT OF SAMPLE SIZE

We investigate whether the number of observations affects the performance of causal discovery methods. One could expect that neural based models, similarly to independence testing ones, will improve when more data is supplied (Kalisch & Bühlmann, 2007). We compare benchmarked methods on dataset with varying number of observational samples, ranging from 20 to 8,000 observations.

The results, presented in Figure 13, reveal no consistent pattern of improvement in the ESHD$_{CPDAG}$ metric as observational sample size increases, despite extensive hyperparameter tuning (as described in Section 3.1). For example, DiBS shows the best performance on larger datasets, but its improvements plateau after around 800 samples. Similarly, BayesDAG shows only marginal improvements with larger sample sizes and is unable to outperform DiBS. DCDI improves up to 250 samples and then maintains consistent performance regardless of the sample size, similar to DiBS. Interestingly, SDCD's performance is poor on datasets with small number of observations but begins to improve once sample sizes exceed 250, though is unable to reach DCDI's performance, for larger sample sizes the rate of improvement decreases.

Further analysis of the effect of sample size on smaller graphs ER(5, 1)is presented in Figure 12 in Appendix C.4. Overall, the results on smaller graphs align with the trends observed on larger graphs. Specifically, while some methods improve with increasing sample size, others show inconsistent or even degraded performance.

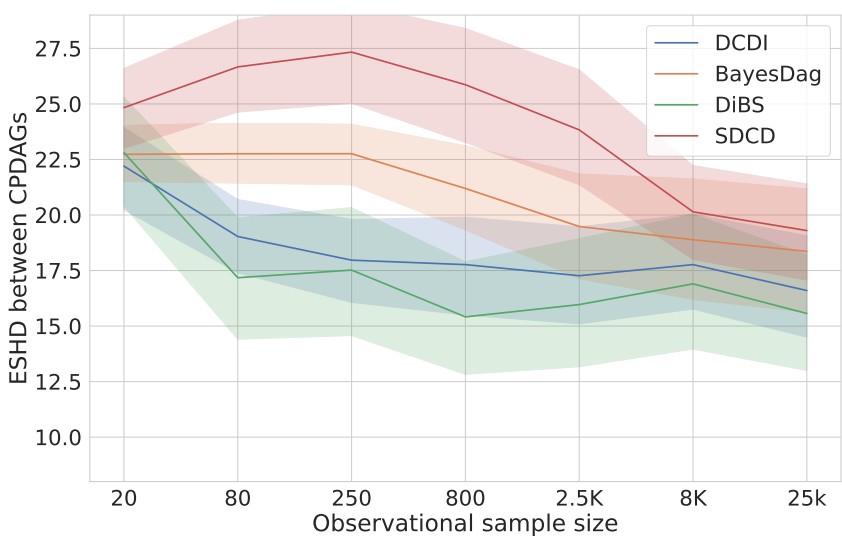

Figure 2: Comparison of ESHD$_{\text{CPDAG}}$ for different methods using the [4, 4] architecture, for ER(10, 2) dataset, averaged over 30 samples.

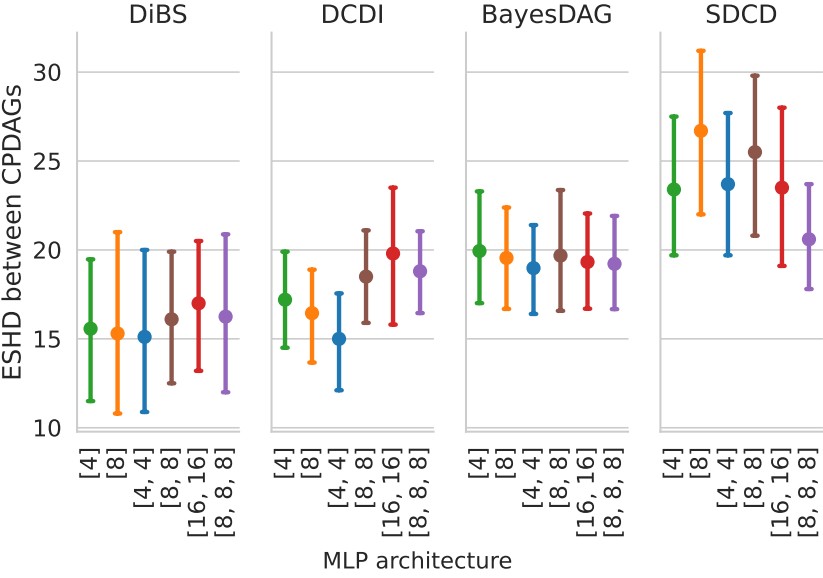

Figure 3: Comparison of ESHD$_{\text{CPDAG}}$ using different MLP architectures as functional approximator for ER(10, 2) dataset and 800 observational samples, averaged over 30 samples.

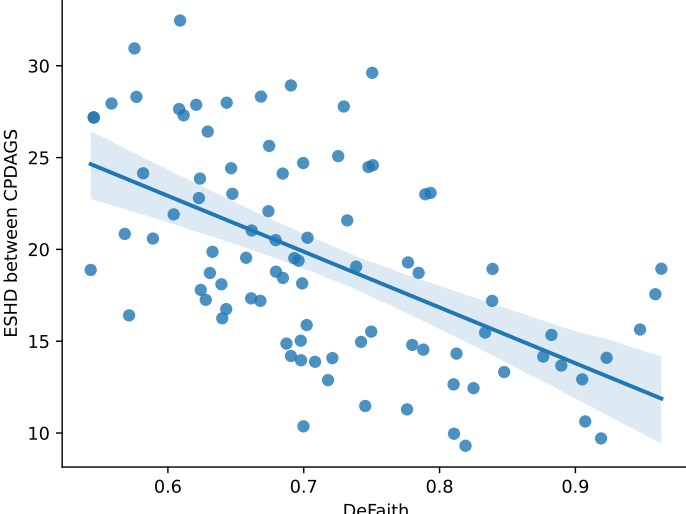

Figure 4: Linear regression fit between the average performance of neural causal discovery methods and faithfulness accuracy measure.

### 3.4 COMPARISON OF NEURAL MODEL ARCHITECTURES

Finally, we investigate the impact of the neural model architecture, used as the functional approximator, on the performance of the benchmarked methods. Specifically, we assess how the capacity of different architectures influences the ability to uncover causal relationships from synthetic data. To provide a comprehensive evaluation, we explored architectures with 1, 2, and 3 layers, configured with 4, 8, and 16 hidden units.

Results, presented in Figure 3 show the comparison of $\text{ESHD}_{\text{CPDAG}}$ metric for the benchmarked architectures across all methods on dataset with 800 samples. We find that the choice of neural architecture has no significant impact on performance across methods. We conclude that any of the tested MLP architectures provides sufficient capacity to model the underlying distribution effectively. Additionally for BayesDAG and SDCD we implemented layer normalization and residual connections. We investigated the impact of this changes in architectures and did not found any significant differences, see Figure 11. The details and additional experimental results are in Appendix C.3.

## 4 MEASURING IMPACT OF FAITHFULNESS VIOLATION

In this section we explore how violations of the faithfulness assumption impact the performance of neural causal discovery methods. In Section 3, we showed that despite various attempts to scale up data and model complexity, the performance of these methods remains stagnant, possibly due to deeper challenges related to the underlying data properties and the limitations inherent to the algorithms. This leads us to investigate whether violations of the faithfulness assumption, common in synthetic non-linear data, might be the key factor limiting performance improvements.

The faithfulness assumption translates into the set of conditional independence statements that all need to be satisfied. As mentioned in Section 2, synthetic non-linear data rarely adheres to faithfulness assumption, rendering binary criterion not practical. To address this, we introduce a degree of faithfulness metric, denoted *DeFaith*, which captures the faithfulness violations on a continuous scale.

Inspired by Zhang & Spirtes (2003), we use Spearman's rank correlation coefficient to quantify the conditional dependencies in the dataset. We define a predictor that classifies nodes as independent if conditional Spearman's rank correlation coefficient computed based on a dataset $D$ is smaller than a certain threshold.

*DeFaith* is the quality of this predictor measured by Area Under Receiver Operator Curve computed over all possible pairs of variables $a, b$ and separation sets $S \subseteq V \setminus \{a, b\}$. Formally,

---

**Algorithm 1** Overview of NN-opt method

---

1: **Input:** Set of nodes $V$, training data $\{D_i\}_{i \in V}$, regularization coefficient $\lambda$, $\mathbb{G}$ the space of DAGs with nodes $V$
2: # *Part 1: Network fitting*
3: **for** $i \in V$ and $\pi \subseteq V \setminus \{i\}$ **do**         $\triangleright$ For each variable and each possible parent set
4:     $\theta_{i,\pi} \leftarrow \text{TRAINNETWORK}(i, D, \pi)$         $\triangleright$ Train ensembles of 3 networks
5: **end for**
6: # *Part 2: Exhaustive graph search*
7: **for** $G \in \mathbb{G}$ **do**         $\triangleright$ Evaluate all possible DAGs
8:     $\text{score}_G \leftarrow \sum_{i \in V} \text{COMPUTENLL}(D_i, D_{Pa_i^G}, \theta_{i,Pa_i^G})$   $\triangleright$ Compute NLL using ensemble
9:     $\text{score}_G \leftarrow \text{score}_G + \lambda \cdot |G|$         $\triangleright$ Add regularizing term
10: **end for**
11: **Output:** $\arg\max\{\text{score}_G : G \in \mathbb{G}\}$

---

$$DeFaith(D, G) = \underset{a,b \in V, S \subseteq V \setminus \{a,b\}}{AUROC} (1 - abs(\rho_s^D(a, b|S)), \mathbf{1}[a \perp_G b|S])$$

where $V$ is set of nodes in graph $G$, $a \perp_G b|S$ denotes $d$-separation between nodes $a$ and $b$ given $S$, and $\rho_s^D(a, b|S)$ denotes conditional Spearman's rank correlation coefficient computed based on dataset $D$. The measure attains a value of 1.0 for faithful distributions.

In this experiment, we generate 30 graphs from the ER(10, 2) class, introduced in Section 3.1. Based on each graph, we define three different SCMs, resulting in 90 distinct distributions. Each dataset consists of 8,000 observational samples. We then evaluate the *DeFaith* of each distribution and compute the performance of the selected neural-based causal discovery methods.

In Figure 4 we present the relationship between average performance of all methods and the degree of faithfulness for all 90 distributions in the dataset. The performance is better (lower SHD) for distributions with higher degree of faithfulness. The Spearman's rank correlation coefficient is $\rho = -0.58$. This result proves the strong anti-monotonicity between the faithfulness accuracy and methods' performance.

## 5 ESTIMATING UPPER BOUND ON PERFORMANCE

In this section, we investigate the limits of the performance of score-based neural causal discovery methods. To do this we develop a method dubbed as NN-opt method, to compute an experimental upper bound on the performance. As for the benchmarked methods, the goal of NN-opt method is to find a structure that minimizes the regularized log-likelihood of data, therefore it is expected to recover a graph from the correct MEC class when the faithfulness assumption holds (see Section 2).

The method overview is in Algorithm 1. It is based on the common approach used by score-based neural causal discovery methods described in Section 2. The procedure consists of two steps. First, we train neural networks to approximate functional relationships between variables. Contrary to benchmarked methods we train a separate network for each parent set instead of training one for all. This renders functional approximation fitting procedure completely independent from structure search. Therefore, it simplifies the training task and allows for strict control of the training procedure via validation loss monitoring. Second, we conduct an exhaustive search over the space of DAGs to find the structure that minimizes the log-likelihood loss. For increased stability of this step, we use an ensemble of 3 neural networks to compute the log-likelihood of the data under various structures.

The approach is exhaustive both in the sense of structure search and in neural network training, trading computational efficiency for additional precision. NN-opt is a brute force technique intended to be able to reach the limits of score-based neural causal discovery approaches. NN-opt is helpful as an upper-bound benchmark but is not practical to use.

We expect the method to improve with the number of samples and stabilize when the data becomes sufficiently large. Therefore, we applied NN-opt method to datasets of various sizes. The results are presented in Figure 5 on the left. For very small datasets we observe rapid improvement in

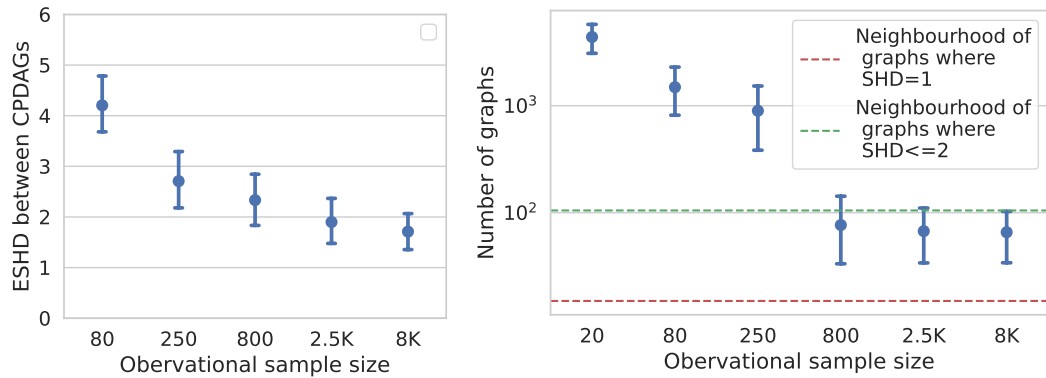

Figure 5: Comparison of the performance of NN-opt method depending on data size (left), and comparision of number of DAGs with score higher than true graph (right). Averaged over 90 samples

terms of ESHD$_{\text{CPDAG}}$, but as the sample size grows, the structure discovery accuracy stabilizes. For sample sizes of 2,500 and 8,000, the value of ESHD$_{\text{CPDAG}}$ is just below 2. In the dataset used for this experiment, the average number of edges in CPDAG is around $8.4$, meaning that on average almost 25% of the edges are predicted incorrectly.

Furthermore, to show that the problem is systematic, we present the number of graphs with a higher score than the ground true DAG in Figure5 on the right. For smaller datasets (with no more than 250 samples) there are around 1000 graphs or more with scores higher than the ground true graph. The number stabilizes around 65 structures, that scored higher than the ground true graph, for bigger datasets. This number is close to the number of graphs with SHD distance $\leq 2$ from the ground truth, depicted by the green line in the figure. These findings demonstrate the methods' consistent inability to identify correct structures.

We argue that this result shows the inherent limitations of the score-based neural causal discovery algorithms due to the violation of the faithfulness assumption. Our NN-opt method controls errors raised from both functional approximations fitting and structure search. Thus violation of faithfulness is the only probable source of errors.

To ensure the validity of the result we performed an extensive hyperparameter search, including models with various architectures. Details of described experiments can be found in Appendix B.

## 6    RELATED WORK

**Causal discovery without the faithfulness assumption**    While many causal discovery methods rely on the faithfulness assumption, alternative conditions have been proposed. One notable approach is the adjacency-faithfulness assumption, introduced by Ramsey et al. (2006) in the conservative PC algorithm. This assumption, which is less restrictive than full faithfulness, leads to more robust with minimal computational overhead. In the context of linear structural causal models (SCMs), Van de Geer & Bühlmann (2013) demonstrated that a sparsity-based assumption can effectively reveal the underlying causal structure. Similarly, Isozaki (2014) proposed a method to reduce unnecessary independence tests during structure discovery, offering greater robustness against violations of faithfulness due to statistical errors. More recently, Ng et al. (2021) suggested another causal discover method, based on relaxed faithfulness assumption that requires less independence tests to be fulfilled. Marx et al. (2021) explores a weaker alternative to the faithfulness assumption, called the 2-faithfulness assumption, and suggests how to construct a causal discovery algorithm based on it. Moreover, Lippe et al. (2022) introduced a neural-based approach that uses interventional data, avoiding the faithfulness assumption altogether.

**Describing faithfulness violations**    Faithfulness violation has been extensively explored in the linear setting by (Uhler et al., 2013). They showed that the conditions that would allow for discovering

the true independencies in a finite sample regime are rarely met when making use of linear synthetic data. Additionally, they proved that the bigger the graph the more difficult it is to find a faithful distribution. Zhang & Spirtes (2003) provided theoretical conditions for violation of faithfulness being detectable during training. More generally, Andersen (2013) described reasons, why faithfulness is likely violated in complex, evolved real-world systems. To the best of our knowledge, we are the first to estimate the limits of the score-based neural causal discovery methods on unfaithful data.

**Benchmarking** There is a multitude of recent benchmarks that use real-world data to assess the performance of causal discovery methods (Chevalley et al., 2022; Mehrjou et al., 2022). However, these datasets lack the ground truth structure rendering structure discovery accuracy assessment impossible. Additionally, these works usually focus on classical, not neural, causal discovery methods. Some recent work is concerned with the quality of evaluations and performance under assumptions violations. Karimi-Mamaghan et al. (2024) investigates metrics for Bayesian causal discovery in a linear setting. Their finding suggests that the standard structure-based metrics do not align well with downstream task performance when structure uncertainty is high (especially for bigger graphs), Montagna et al. (2023) evaluates classical causal discovery methods under different assumption violations. In our work, we focus on a unified, synthetic, and challenging setup to thoroughly evaluate neural causal discovery claims of being general and accurate. Most recently, Zhou et al. (2024) introduced a comprehensive benchmark, but they did not compare neural-based methods in their work.

## 7 LIMITATIONS & FUTURE WORK

- Work of Lippe et al. (2022) suggests that interventional data can replace the need for faithfulness assumption. A valuable extension of our research would be to evaluate the performance of the benchmarked methods on interventional datasets to understand their limitations and potential improvements in this context.

- Our work provides experimental evidence for the scale of the impact of violation of faithfulness on performance in a challenging non-linear setting. It would be beneficial for the community if some theoretical results (akin Uhler et al. (2013); Zhang & Spirtes (2003) were derived in a non-linear setting.

- While our, experimental upper bound, NN-opt method is based on common, with benchmarked methods, theoretical principles. We leave strict theoretical justification of its optimality for future work.

- In this work we present the method that allows to estimate the upper bound on performance of score-based neural causal discovery methods on any dataset and provide numerical results for the Erodos-Renyi class of graphs. The results could be computed for more classes and even some small real-world or real-world inspired graphs, see Elidan (2001).

## 8 CONCLUSIONS

In this work, we present compelling evidence that the faithfulness assumption is a major limiting factor in advancing causal discovery. Our findings demonstrate that the accuracy of structure recovery is correlated with the degree of faithfulness violation. Additionally, we introduce a novel method to calculate the upper bound of performance for score-based neural causal discovery methods, revealing serious limitations. Our results highlight the need for a paradigm shift. We argue that further progress in causal discovery requires moving beyond the faithfulness assumption and encourage researchers to explore alternative conditions. The implications of our work extend beyond theoretical advancements. By challenging the faithfulness assumption, we open up avenues for more robust and generalizable methods in causal discovery, which could have far-reaching consequences in fields like healthcare, economics, and policy-making.

## 9 REPRODUCIBILITY STATMENT

We put effort and resources to ensure that presented experiments can be reproduced by the research community. Specifically, we provide detailed descriptions of the data generation process, benchmarking score-based neural causal discovery methods and proposed NN-opt method.

Our dataset generation process is based on code included in DCDI code repository (Brouillard et al., 2020) and the details of this processed can be found in Section 3.1 and Appendix C.1. The performance of the selected causal discovery methods, for the benchmark, was compute using official repositories released by authors: DCDI (Brouillard et al., 2020), SDCD (Nazaret et al., 2024), BayesDag (Annadani et al., 2023) and DiBS (Lorch et al., 2021). The range of tested hyperparameters and the selected values can be found in Section 3.1 and Appendix C.2.

The description on NN-opt method is provided in Section 5 moreover the high level overview of method is in Algorithm1. Hyperparameter selection is described in Appendix B.

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

## A  ADDITIONAL BACKGROUND INFORMATION

### A.1  $d$-SEPARATION

Two nodes $A$ and $B$ in a DAG are said to be $d$-**separated** by a set of nodes $Z$ if all paths between $A$ and $B$ are blocked when conditioning on $Z$. A path is considered blocked under the following conditions:

- If a path includes a non-collider node (a node where arrows do not converge, i.e., a chain or fork), conditioning on that node blocks the path. For example, if $A \to C \to B$, or $A \leftarrow C \to B$, conditioning on $C$ makes $A$ and $B$ independent.

- If the path includes a collider (a node where arrows converge, i.e., $A \to C \leftarrow B$), the path is blocked unless either the collider itself or one of its descendants is conditioned on. For instance, in the path $A \to C \leftarrow B$, conditioning on $C$ or its descendants would unblock the path, making $A$ and $B$ dependent.

- If there are multiple paths connecting $A$ and $B$, all paths must be blocked for $A$ and $B$ to be considered $d$-separated. Even if one path remains unblocked, $A$ and $B$ are d-connected, meaning they are dependent.

In causal discovery, we are interested in making statements about the relationship between the causal graph and the data distribution. Given a causal graph $G$ and the data distribution $P$, the **Markov assumption** states that if variables $A$ and $B$ are $d$-separated in the graph $G$ by some conditioning set $C$, then $A$ and $B$ are conditionally independent in the distribution $P$ when conditioned on the same conditioning set $C$. Formally, this can be written as:

$$A \perp\!\!\!\perp_G B | C \Rightarrow A \perp\!\!\!\perp_P B | C \tag{6}$$

### A.2  EXAMPLE OF FAITHFULNESS VIOLATION

In this subsection we will illustrate a faithfulness violation for a simple 3 nodes structural causal model with linear functions and additive Gaussian noise. Such a setup is aimed at showing example of faithfulness violation while maintaining simplicity. The example and graphics is from (Uhler et al., 2013).

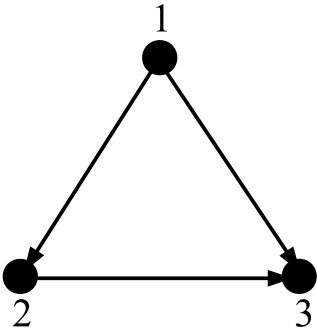

Figure 6: Simple 3 nodes graph $G$.

First lets define a structural causal model on a graph $G$ shown in graph 6.

$$\begin{aligned}
X_1 &= \varepsilon_1, \\
X_2 &= a_{12}X_1 + \varepsilon_2, \\
X_3 &= a_{13}X_1 + a_{23}X_2 + \varepsilon_3, \\
(\varepsilon_1, \varepsilon_2, \varepsilon_3) &\sim \mathcal{N}(0, I),
\end{aligned}$$

Since data is linear we can use covariance to measure dependency of variables. Using defined structural causal model, we can write:

$$\text{cov}(X_1, X_2) = a_{12}, \tag{7}$$
$$\text{cov}(X_1, X_3) = a_{13} + a_{12}a_{23}, \tag{8}$$
$$\text{cov}(X_2, X_3) = a_{12}^2 a_{23} + a_{12}a_{13} + a_{23}, \tag{9}$$
$$\text{cov}(X_1, X_2 \mid X_3) = a_{13}a_{23} - a_{12}, \tag{10}$$
$$\text{cov}(X_1, X_3 \mid X_2) = -a_{13}, \tag{11}$$
$$\text{cov}(X_2, X_3 \mid X_1) = -a_{23}. \tag{12}$$

If we define $a_{13}, a_{23}, a_{12}$ in such a way that:

$$a_{13} * a_{23} - a_{1,2} = 0$$

then we get a situation where: nodes 1 and 2 are not d-separated given node 3 in a graph $G$ and $X_1 \perp\!\!\!\perp X_2 | X_3$ which is a violation of faithfulness.

## B  NN-OPT METHOD DETAILS

**Details of experiments with NN-opt method**   In order to test which architecture perform best, we conducted an experiment, training NN-opt method with different sizes of neural networks. The trained models were judged in terms of negative log likelihood and their performance on the task of causal discovery measured as $\text{ESHD}_{\text{CPDAG}}$. For each tested architecture, we performed the search for the best regularization coefficient, the tested coefficients were: $[0.1, 0.3, 1.0]$. Among all models, the best results were consistently obtained for regularization coefficient = 0.3. The learning rate was set to 0.0003. The results of the experiments are shown in Table 2. As we can see, the best , both in case of NLL and $\text{ESHD}_{\text{CPDAG}}$ was model with two layers and hidden dimension of size 8. Notably this is the same architecture, as was used to generate data.

**Selected hyperparameters:** Number of layers = 2, hidden dimension = 8, regularization coefficient = 0.3.

| Model architecture | NLL | $\text{ESHD}_{\text{CPDAG}}$ |
|---|---|---|
| [4] | $0.33_{(0.22, 0.43)}$ | $3.63_{(2.83, 4.67)}$ |
| [4, 4] | $0.2_{(0.1, 0.3)}$ | $3.15_{(2.0, 4.65)}$ |
| [4, 4, 4] | $0.23_{(0.14, 0.34)}$ | $3.03_{(2.33, 4.07)}$ |
| [8] | $0.18_{(0.06, 0.29)}$ | $2.13_{(1.43, 3.07)}$ |
| [8, 8] | $\mathbf{0.13}_{(0.02, 0.24)}$ | $\mathbf{1.23}_{(0.77, 1.87)}$ |
| [8, 8, 8] | $0.22_{(0.12, 0.32)}$ | $2.77_{(1.97, 3.67)}$ |
| [16] | $0.14_{(0.03, 0.26)}$ | $1.77_{(1.1, 2.73)}$ |
| [16, 16] | $0.33_{(0.24, 0.42)}$ | $2.4_{(1.0, 4.32)}$ |
| [16, 16, 16] | $0.88_{(0.8, 1.0)}$ | $4.0_{(3.07, 4.97)}$ |

Table 2: The performance of NN-opt method models with different architectures. The numbers in the subscripts, correspond to 0.95 confidence intervals. The experiments were performed on 30 graphs.

## C  DETAILS ABOUT BENCHMARK AND EXTENSIONS

### C.1  DATASET GENERATION DETAILS

The data is generated using a fully connected MLP with two hidden layers of 8 units each, initialized with random weights drawn from a uniform distribution and use the ReLU (Nair & Hinton, 2010) activation function to introduce non-linearity. The neural network models the relationships between variables in the underlying DAG, where each node represents a variable and the edges capture dependencies between these variables. The input variables, which serve as the initial causes in the graph, are sampled from normal distributions. The noise added to the system is sampled from a Gaussian distribution $\mathcal{N}(0, 0.1^2)$, simulating uncertainty in the model. The dataset consists of 100,000 data points, and the data is rescaled to maintain consistency across samples.

## C.2 MODEL HYPERPARAMETERS

We performed extensive hyperparameter tuning for all methods. In addition to the MLP architecture grids described in Appendix C.3, the following hyperparameter grids were explored:

**DCDI** **Grid search**: Regularization coefficients tested: [0.1, 0.3, 1, 2]. Values below 0.001 or above 5 led to poor performance. **Selected**: Regularization coefficient = 1, learning rate = 0.001, Augmented Lagrangian tolerance = $10^{-8}$.

**DiBS** **Grid search**: Alpha linear: [0.01, 0.02, 0.05], kernel parameters: h latent: [0.5, 1.0, 2.0], h theta: [20.0, 50.0, 200.0], step size:[0.05, 0.03, 0.01, 0.005, 0.003]. **Selected**: Alpha linear = 0.02, h latent = 1.0, h theta = 50.0, step size = 0.03.

**BayesDAG** **Grid search**: Scale noise: [0.1, 0.01], scale noise p: [0.1, 0.01, 1.0], lambda sparse: [50.0, 100.0, 300.0, 500.0]. **Selected**: Scale noise = 0.1, scale noise p = 0.01, lambda sparse = 500.0.

**SDCD** **Grid search**: Constraint modes: ["exp", "spectral radius", "matrix power"]. The $\text{ESHD}_{\text{CPDAG}}$ metric showed similar results across modes. **Selected**: Spectral radius was chosen for faster computation, with a learning rate of 0.0003.

For each of these method, all other parameters were retained from the original paper or code.

## C.3 MODEL ARCHITECTURE COMPARISION WITHIN METHOD

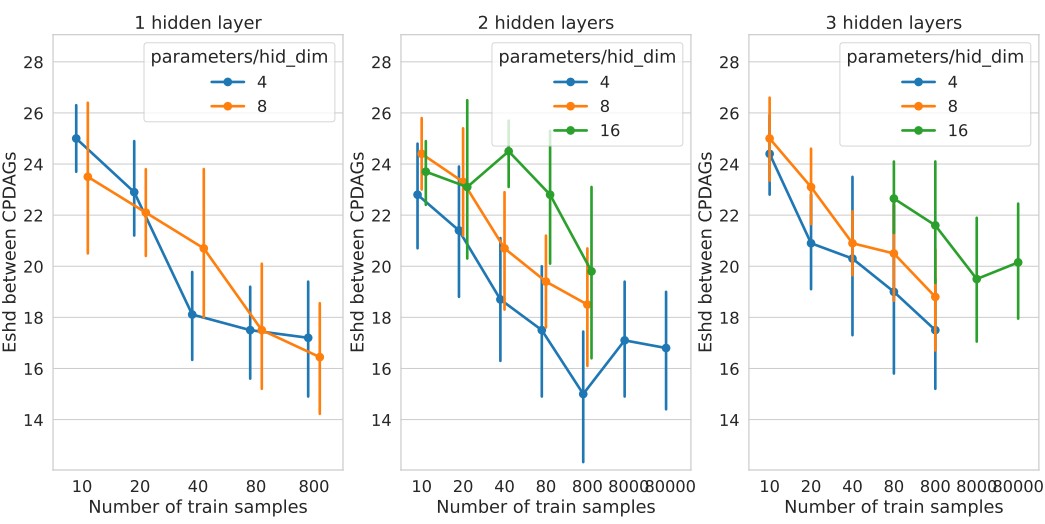

Figure 7: Comparison of the $\text{ESHD}_{\text{CPDAG}}$ of DCDI for datasets with different observational sample size. The result is based on 10 graphs.

**DCDI** In Figure 7, we present the performance analysis of the DCDI across various neural network configurations. Our results reveal that the optimal performance is generally achieved by a two-layer model with a hidden dimension of 4. Interestingly, we observe that more expressive models exhibit diminished performance relative to the smaller models.

**DiBS** Figure 8 presents the performance analysis of the DiBS method across various neural network configurations. As with the DCDI method, we evaluate models with different numbers of layers and hidden dimension sizes. Consistent with DCDI, we find that the optimal performance for DiBS is achieved by a two-layer model with a hidden dimension of 4. However, the performance landscape for DiBS exhibits less variability across different model configurations. Single-layer models perform nearly as well as the optimal two-layer model.

Furthermore, we observe that more expressive models do not show a significant degradation in performance as was seen with DCDI. The overall differences in metric across all tested configurations are relatively small for DiBS, indicating a more consistent performance across varying levels of model complexity.

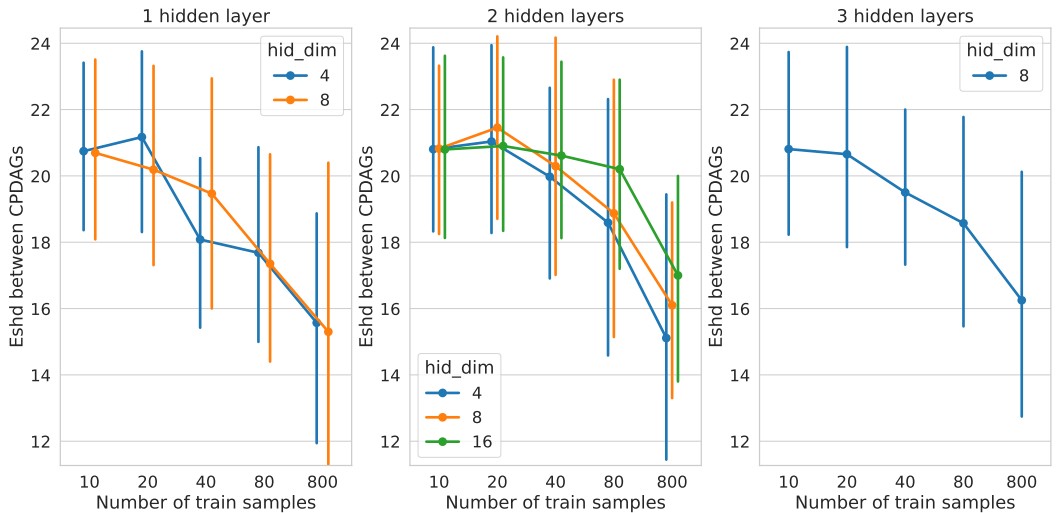

Figure 8: Comparison of the performance of DiBS depending on the model architecture and number of samples.

**BayesDAG** Figure 9 compares the performance of BayesDAG across different model architectures and sample sizes. For smaller sample sizes, BayesDAG's performance remains consistent, with noticeable differences emerging only at a sample size of 800. This suggests that BayesDAG requires more data to fully leverage its model capacity, unlike what we observed for DCDI and DiBS, where performance varied more significantly across sample sizes. Notably, the best-performing architecture for DiBS is a two-layer MLP with a hidden dimension of 4.

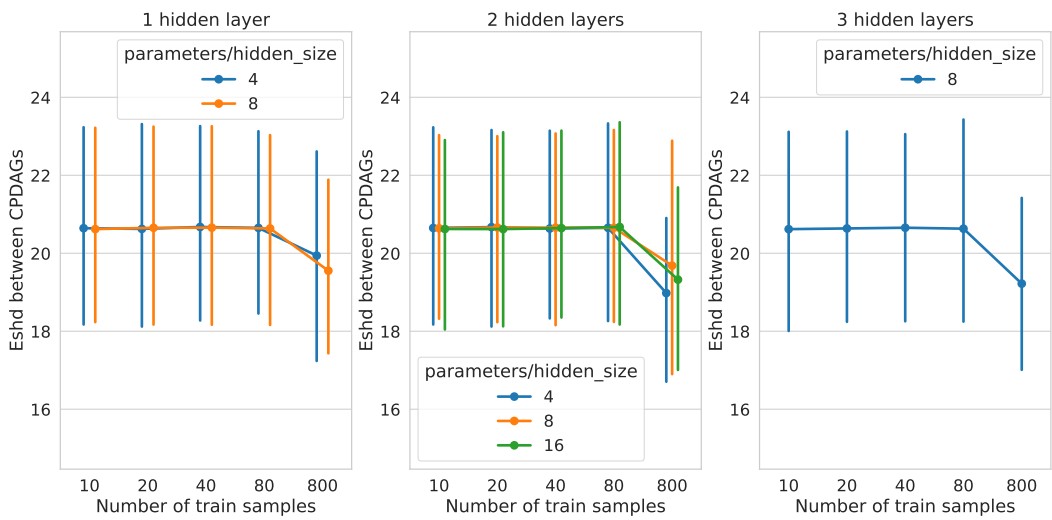

Figure 9: Comparison of the performance of DiBS depending on the model architecture and number of samples.

**SDCD** Figure 10 presents a similar comparison of SDCD performance across different MLP architectures and sample sizes. Interestingly, the three-layer architectures show stagnant performance regardless of sample size, while the one-layer models exhibit significant improvement as the sample size increases. Overall, the best performance is achieved with a one-layer MLP with 8 hidden units, although it remains comparable to the one-layer MLP with 4 hidden units and the two-layer MLP with 4 hidden units.

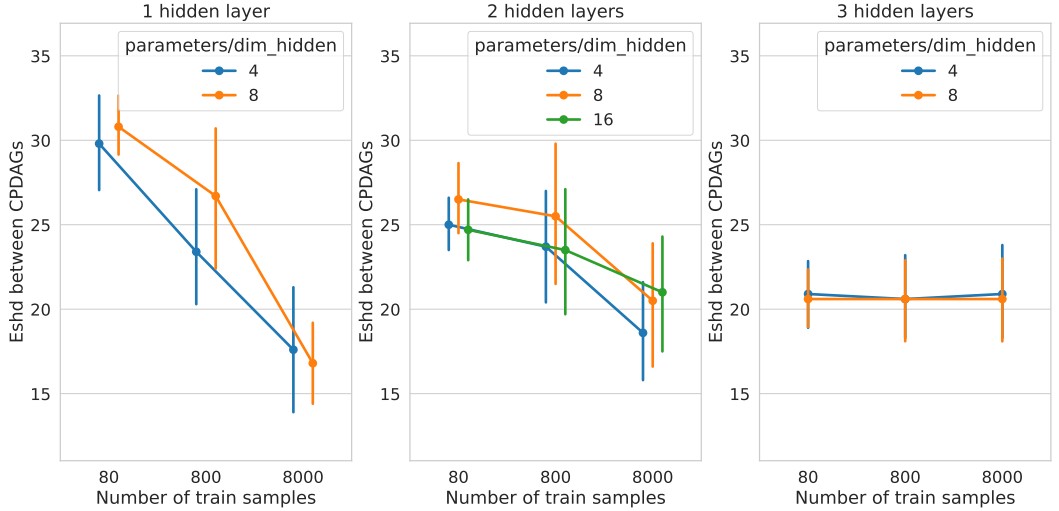

Figure 10: Comparison of the performance of SDCD depending on the model architecture and number of samples.

**Model architecture** Inspired by BayesDAG, we also implemented layer normalization and residual connections to assess their impact. We conducted additional experiments on both the best-performing model ([4, 4]) and the largest model ([8, 8, 8]). The size of networks was similar to the one proposed in articles introducing tested methods: in DCDI it was [16, 16], for SDCD it was [10, 10], for DiBS [5, 5] and for BayesDAG it was a two layer network with a hidden size varying with dimensionality. The results of these tests are presented Figure 11. We show, there is no significant and consistent improvement across all networks, supporting our initial conclusion that variations in MLP architecture have minimal impact on performance.

### C.4 INFLUENCE OF SAMPLE SAMPLES ON PERFORMANCE ON THE GRAPH WITH ER(5, 1)

Figure 12 shows the ESHD$_{CPDAG}$ of benchmachmarked methods for different sample sizes. For all observational sample sizes, SDCD and DCDI have a large confidence interval. For datasets with 2,500 and 8,000 samples, BayesDAG performs better than other benchmarked methods, getting small confidence interval for 8,000 samples.

### C.5 ADDITIONAL RESULTS FOR SDCD AND DIBS

## D JUSTIFICATION OF EVALUATION METRICS

We design metrics based on popular SHD, F1-score metrics, which we explain shortly below.

**The Structural Hamming Distance.** SHD (Tsamardinos et al., 2006) quantifies the difference between the predicted graph and the ground truth graph by counting the number of edge insertions, deletions, and reversals required to transform one into the other. SHD values indicate the degree of error in recovering the true causal structure: lower SHD values signify better predictions, while higher values indicate more significant discrepancies.

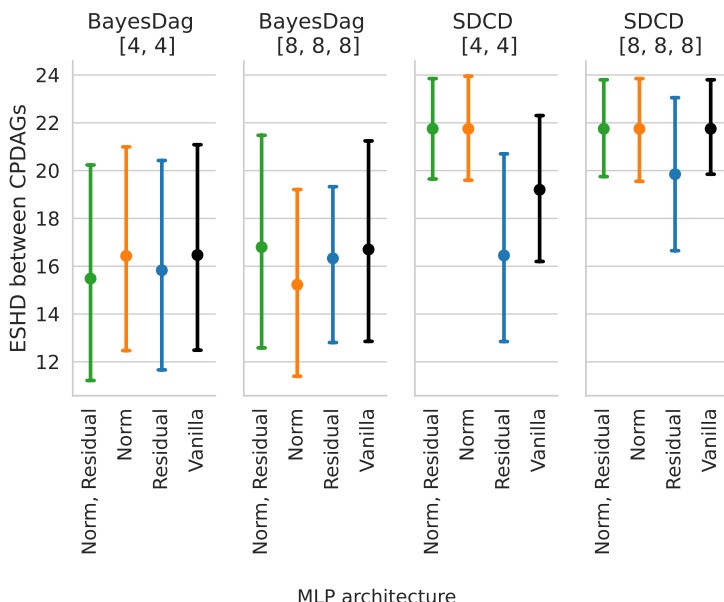

Figure 11: Comparison of the performance of SDCD depending on the model architecture and number of samples.

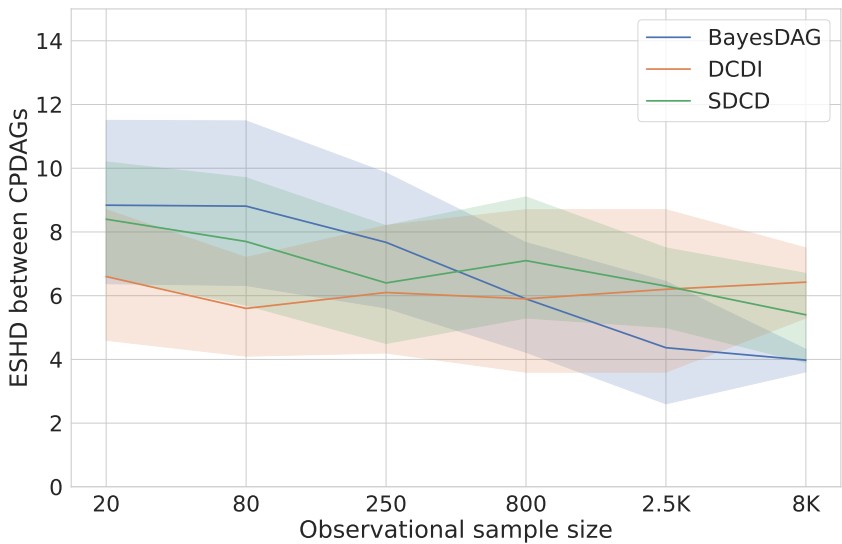

Figure 12: Comparision of $\text{ESHD}_{\text{CPDAG}}$ for benchmarked methods on ER(5, 1) dataset, averaged over 10 graphs.

**The F1-score.** The F1-Score measures the harmonic mean of precision and recall for edge predictions, where precision reflects the fraction of correctly predicted edges among all predicted edges, and recall reflects the fraction of correctly predicted edges among the true edges.

We evaluate causal discovery methods based on observational data. In general, in this setup, it is only possible to recover true DAG up to a Markov Equivalence Class, a class of graphs with the same conditional independence relationships, due to identifiability issues TODO cite pearl?. If we were to compare the predicted and ground true graphs using standard metrics like SHD or F1-score we would obtain distorted results — graphs from the MEC class do not generally receive these metrics' optimal values.

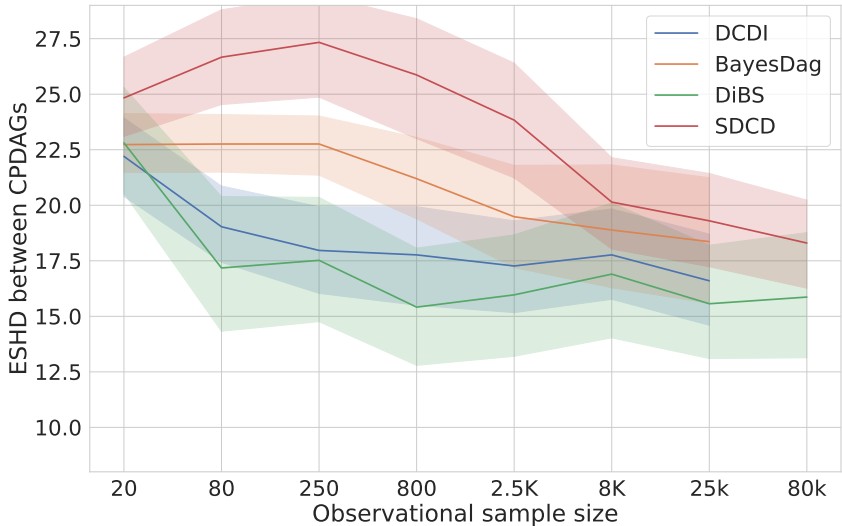

Figure 13: Comparison of ESHD$_{\text{CPDAG}}$ for different methods using the [4, 4] architecture, for ER(10, 2) dataset, averaged over 30 samples.

Therefore, we modify the formulation of the metrics to account for the limitations of causal discovery from observational data. We define ESHD$_{\text{CPDAG}}$ and F1-Score$_{\text{CPDAG}}$. These metrics attain their optimal values, 0 and 1 correspondingly, for all DAG from ground truth MEC. Additionally, some of the benchmarked methods are Bayesian thus return the posterior over possible solutions. For those methods, we design metrics that compute the expected value over the posterior and approximate it with the Montecarlo estimator based on a sample of size 100.

We define **Expected SHD between CPDAGs** as:

$$\text{ESHD}_{\text{CPDAG}}(\mathcal{G}, \mathbb{G}) = \mathbb{E}_{\mathcal{G}^* \sim \mathbb{G}}[\text{SHD}(\text{CPDAG}(\mathcal{G}), \text{CPDAG}(\mathcal{G}^*))], \tag{13}$$

where $\mathbb{G}$ is the resulting distribution of graphs, $\mathcal{G}^*$ is a graph sampled from $\mathbb{G}$ and $\mathcal{G}$ is the ground true graph. Similarly, we compute the **Expected F1-Score between the CPDAGs**:

$$\text{F1-Score}_{\text{CPDAG}}(\mathcal{G}, \mathbb{G}) = \mathbb{E}_{\mathcal{G}^* \sim \mathbb{G}}[\text{F1-Score}(\text{CPDAG}(\mathcal{G}), \text{CPDAG}(\mathcal{G}^*))]. \tag{14}$$

## E  JUSTIFICATION OF THE SELECTION OF METHODS

During the preliminary phase, we considered the following methods NO-TEARS (Zheng et al., 2018), NO-BEARS (Lee et al., 2019), NO-CURL (Yu et al., 2021), GRAN-DAG (Lachapelle et al., 2019), SCORE (Rolland et al., 2022), DAGMA (Bello et al., 2022), DCDFG (Lopez et al., 2022), DCDI (Brouillard et al., 2020), DiBS (Lorch et al., 2021), BayesDAG (Annadani et al., 2023), SDCD (Nazaret et al., 2024), from which we chose DCDI, SDCD, DiBS and BayesDAG. Below we explain why the included ones cover non-included methods.

NO-TEARS is the first method to use augmented Lagrangian and differentiable constraints to enforce DAGness. However, the suggested formulation entangles functional and structural parameters, making NO-TEARS applicable only to linear models or restricted neural networks. The NO-TEARS method was improved in GRAN-DAG (introduces separate adjacency matrix and sampling based on Gumbel softmax) and then in DCDI (accounts for interventional data). We chose to use DCDI as it is the most developed method in this line of work and has clean implementation.

An interesting line of work shows articles introducing methods such as NO-BEARS and DAGMA, that were focused on improving the acyclicity constraint introduced in NO-TEARS, all proposed constraints were unified in the SDCD paper, and a new constraint was proposed, that was shown to

perform the best. Additionally, SDCD is compared against SCORE and DCDFG again presenting better performance.

The two other methods are from the class of Bayesian approaches. DiBS method is selected as a Bayesian approach that uses classic NO-TEARS-based regularization embedded in its prior. The BayesDAG is based on the NO-CURL parametrization of DAGs and provides improvements to the optimization pipeline (uses MCMC instead of SVGD).

We argue that this selection of four methods summarizes various research directions and improvements explored in neural causal discovery over the last four years and well represents the spectrum of existing approaches.

## F  EXPERIMENTS ON REAL-WORLD STRUCTURES

To further substantiate our findings, we conducted additional experiments using Bayesian network structures sourced from the bnlearn repository (Elidan, 2001). This repository provides networks that represent real-world systems. However, the functional relationships in these networks are often limited to simple models, such as linear Gaussian or discrete distributions. To address this limitation, we utilized the graph structures from bnlearn but generated functional relationships consistent with those used in the synthetic benchmark.

We selected cancer and sachs structures. cancer has 5 nodes and 4 edges and we employed the set of hyperparameters that yielded the best performance for the ER(5, 1) for each method. sachs has 11 nodes and 17 edges and we employed the set of hyperparameters that yielded the best performance for the ER(10, 2) for each method.

The results, presented in Figure 14, align with the observations detailed in Section 3. Across all methods, we observed either consistently poor performance regardless of sample size or very slow improvements, exhibiting diminishing returns with increasing data.

Most methods significantly underperformed compared to the NNOpt approach. An exception is DiBS, which achieved results comparable to the upper bound on the cancer graph, a behavior similar to its performance on the ER(5, 1) class of graphs.

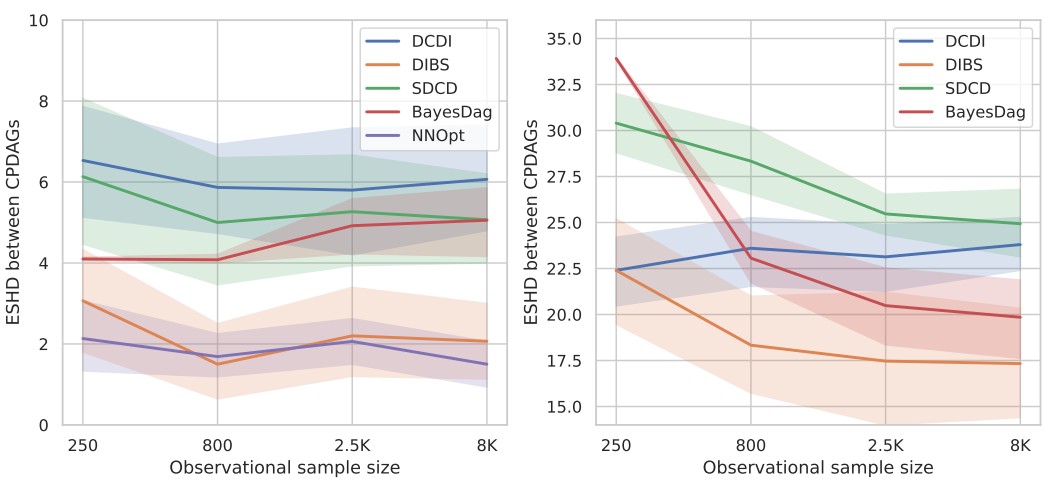

Figure 14: *On the left:* Results of the benchmark on cancer structure. *On the right:* Results of the benchmark on sachs structure. In both plots, the 95% bootstrap confidence interval is provided as a shaded area. The results are computed on 15 distributions.

