# OpenReview forum: "Since Faithfulness Fails: The Performance Limits of Neural Causal Discovery"
_ICLR.cc/2025/Conference — Submitted to ICLR 2025_

### Official Review · Reviewer_CV1a · 2024-10-31

**Soundness:** 2
**Presentation:** 3
**Contribution:** 2
**Rating:** 6
**Confidence:** 4

**Summary:**

The paper presents the idea that the ''faithfulness assumption" is holding back the improvements in the neural causal discovery research. The work starts with empirically demonstrating the strenghts and limitations of neural causal discovery. To this end, a benchmark is proposed where several of the discovery algorithms are evaluated.  Then a degree of faithfulness metric is proposed that is helpful to estimate to what degree does faithfulness effect the discovery algorithms.

**Strengths:**

1. The premise is really interesting. Identifying the faithfulness assumption as the main culprit is an important hypothesis that can have wide ranging impact.

2. The paper is relatively well written.

**Weaknesses:**

There are several issues with the paper that forces me to go with a lower rating.

1. Although the premise is interesting, the overall work does not justify the premise. For example, using a AUC-ROC curve to measure faithfulness is not very well justified. Also quantifying the conditional dependecies by a correlation coefficient is not very innovative/interesting.

2. The works talks about why such assumptions are a problem for real world datasets, but the experimental evaluations are on synthetic data sets. This kind of defeats the message.

3. The NN-opt method seems to pretty expensive since all possible DAG's are being evaluated.

4. In Fig 2, SDCD and Bayes DAF do show a downward trend so maybe more evaluations are required before claiming that there is no improvement in the ESHD metric.

5. Spell check required. Eg: Masuring -> Measuring in heading of Section 4.

**Questions:**

Please look at the weaknesses section. Furthermore,

1. Were there any ablation studies conducted for the neural networks as functional aprroximators?

2. I do not see the point of Fig 1 to be honest. It gives no specific information.

---

> ### Author Response · Authors · 2024-11-21
> **Response to Reviewer CV1a**
>
> We thank the Reviewer for providing their review. We are happy to learn that the Reviewer found the research hypothesis interesting and that the paper is cleanly written and well presented. Below we answer specific questions:
>
> **Weaknesses:**
>
> **Ad 1**: Please refer to the general response for a detailed explanation and justification of the DeFaith measure. We also adjusted the description of the DeFaith measure in section 4 accordingly.
>
> **Ad 2**: We use synthetic datasets for practical reasons. It allows us to have full knowledge of various properties of datasets, especially the ground true graph, which are difficult to obtain for real-world datasets. We argue that our results are representative of many real-world problems since they are based on very general set of structures and functional approximators. Additionally, this type of evaluation is very popular in the literature (used in DCDI, NOTEARS, SDCD and many others).
>
> **Ad 3**: To clarify, NN-opt is a brute force technique intended to be as powerful as possible in causal discovery. The price to pay is efficiency. NN-opt is helpful as an upper-bound benchmark but is not practical to use. Please see also the discussion in the general response.
>
> **Ad 4**: We are currently running the requested experiment, we will provide the results as soon as possible.
>
> **Ad. 5**: Thank you for pointing this out. We fixed that in the revised version of the paper.
>
> **Questions:**
>
> **Ad 1**: Yes, please see Appendix C.3 for a comparison of various architectures.
>
> **Ad 2**: We added specific values that summarize our main result - limitations of neural score-based methods on the ER(5,1) graph.
>
> We greatly appreciate the Reviewer's feedback and the opportunity to respond to their comments and questions. We have carefully considered the feedback and provided detailed explanations and evidence to address any concerns or uncertainties. If the Reviewer has any more questions, we are happy to provide any necessary explanation.

---

> > ### Comment · Reviewer_CV1a · 2024-11-24
> > **Response to the rebuttal**
> >
> > I would like to thank the authors for their response. Several concerns remain:
> >
> > 1. I agree that getting the ground graph for real-world tasks is very difficult but then your motivation seems to be not very strong and should be refined, as I mentioned in my initial review.
> >
> > 2. No experimental results yet for point 4.
> >
> > In light of the response, I will increase my score to a 5 but I think the paper needs more work in order to be accepted.

---

> > > ### Author Response · Authors · 2024-11-28
> > > **Response to Reviewer CV1a**
> > >
> > > We thank the Reviewer for engaging in the discussion, raising the score, and providing constructive feedback. We have provided a revised version of the paper with additional experimental results.
> > >
> > > **Ad 1**. To meet the Reviewer's expectations, we conducted an experiment based on real-world structures sourced from the bnlearn repository. This repository provides Bayesian networks that represent real-world systems. However, the functional relationships in these networks are often limited to simple models, such as linear Gaussian or discrete distributions. To address this limitation, we utilized the graph structures from bnlearn but generated functional relationships consistent with those used in the synthetic benchmark. A detailed description of this experiment is provided in Appendix F. The results, presented in Figure 14 in Appendix F, align with the observations on synthetic graphs detailed in Section 3.
> > >
> > >
> > > **Ad 2.** We have conducted additional experiments using larger sample sizes to address the Reviewer's concerns. Specifically, we tested all methods with 25k samples and observed a slight improvement for SDCD, while BayesDAG showed no significant improvement. We have updated Figure 2 in Section 3 of the paper to reflect these results. Furthermore, we extended our experiments to 80k samples for DiBS and SDCD. At this scale, DiBS demonstrated no improvement, while SDCD showed a slight improvement.
> > >
> > >
> > > However, SDCD's performance remains below that of DCDI, and the rate of improvement diminishes as the sample size increases. These additional results confirm our initial conclusions. A plot with the full set of results is included in Figure 13 in Appendix C.5.
> > >
> > >
> > > We would like to emphasize that 80k samples represent a very large dataset, and increasing the sample size further is likely unrealistic in practical applications.
> > >
> > > ***
> > > Thanks to the Reviewer's suggestions, we have significantly improved the experimental section of the paper. We hope we have addressed all the Reviewer's concerns. If so, we kindly ask the Reviewer to consider further increasing the score.

---

> > > > ### Comment · Reviewer_CV1a · 2024-11-28
> > > > **Response**
> > > >
> > > > Thank you for the extra set of experiments. They make the paper more clear.
> > > > 80k samples might be a lot for causal literature but is not a lot in real world examples. Anyways, in light of the changes made, I am revising my rating further.

---

### Official Review · Reviewer_yCC2 · 2024-11-03

**Soundness:** 3
**Presentation:** 4
**Contribution:** 3
**Rating:** 5
**Confidence:** 3

**Summary:**

This paper has claimed that progress in causal discovery requires moving beyond the faithfulness assumption. The experiments shows that the violation of faithfulness would degrade performance. To clearly quantify this phenomenon, a metric called DeFaith has been proposed. Also, as the increase of sample size, the performance of different methods, including DCDI, BayseDAG, DiBS, SCCD, don’t show improvement or even show degraded performance. To investigate the limits of different socre-based neural causal discovery methods, an algorithm called NN-opt has been proposed which consists two step. Through NN-opt, the upper bound of the neural method could be estimated.

**Strengths:**

1. This paper touches some important theoretical aspects of neural causal discovery.
2. Several ordinary causal approaches are compared.

**Weaknesses:**

1. The theoretical analysis can be built more in depth to justify the arguements of the paper.

**Questions:**

Q1:In the structure evaluation, Expected SHD between CPDAGs and Expected F1-Score between CPDAGs have been defined. Compared to other metrics such as SHD and SID, is there any strength or reason to use them?

Q2:DeFaith is proposed to evaluated the faithfulness, but could this metric accurately describe the change of the faithfulness? Could you give more theoretical analysis about the reality between the faithfulness and DeFaith? It is better if this quantification can be justified to be more theoretically "close" to the faithfulness.

Q3:As mentioned before, NN-opt could compute the upper bound of specific methods. In step 2, which is exhaustive graph search, it is not feasible in practice. Is there any way to improve the method to make it practiced?

Q4:In this paper, several methods have been evaluated to search the upper bound of these methods using different sample size and neural structures. From the aspect of experiments, the methods evaluated in the paper may only include some typical ones among all methods and it’s more persuasive to compare more socre-based neural causal discovery methods. Could you give some reason of the selection of the four methods?

---

> ### Author Response · Authors · 2024-11-21
> **Response to Reviewer yCC2**
>
> We thank the Reviewer for taking the time to provide thoughtful and constructive feedback on our submission. We are glad to hear that the Reviewer acknowledges the importance of the presented research topic and that our work is well-presented and sound. Below we provide answers to specific questions raised by the Reviewer.
>
> **Ad Weakness 1**: We agree that theoretical analysis is always useful. However, due to a challenging topic, we first provide experimental evidence supporting our hypothesis. We would be happy to develop a theoretical explanation in the future. In the spirit of answering the Reviewer’s second question, we provide additional justification for the DeFaith metric in the general response.
>
> **Ad Q1**: We evaluate causal discovery methods based on observational data. In general, due to identifiability issues, in this setup, it is only possible to recover true DAG up to a Markov Equivalence Class, a class of graphs with the same conditional independence relationships. If we were to compare the predicted and ground true graphs using standard metrics like SHD or F1-score we would obtain distorted results - graphs from the MEC class do not generally receive these metrics' optimal values. Therefore, we modify the formulation of the metrics to account for the limitations of causal discovery from observational data. The modified metrics attain their optimal values for all DAGs from the ground truth MEC. We added this explanation in Appendix D of the revised paper.
>
> **Ad Q2**: Thank you for asking this question. Please find an additional explanation on how the DeFaith measure relates to the faithfulness assumption in the general response. We also adjusted the explanation in Section 4 accordingly.
>
> **Ad Q3**: To clarify, NN-opt is a brute force technique intended to be as powerful as possible in causal discovery. The price to pay is efficiency. NN-opt is helpful as an upper-bound benchmark but is not practical to use. Please see also the discussion in the general response.
>
> **Ad Q4**: We argue that this selection of four methods summarizes various research directions and improvements explored in neural causal discovery over the last four years and well represents the following selection of methods: NO-TEARS, NO-BEARS, NO-CURL, SCORE, DAGMA, DCDFG, DCDI, DiBS, BayesDAG, SDCD. Please refer to the general response for a detailed explanation. If the Reviewer thinks this comparison is too narrow, we kindly ask the Reviewer to specifically name additional methods that they think it is essential to compare against.
>
> Please let us know if the above answers the Reviewer’s comments and questions. We would also be more than happy to address any other suggestions. In case there are none, and given the Reviewer’s positive outlook on soundness, presentation, and contribution, we would gently ask the Reviewer to consider increasing the score.

---

> > ### Author Response · Authors · 2024-11-30
> >
> > We sincerely appreciate the time and effort the Reviewer has dedicated to reviewing our manuscript. In response to the Reviewer’s questions and concerns, we have provided detailed answers and made corresponding improvements to the manuscript based on their feedback. Please let us know if there are any further comments or concerns regarding our rebuttal or the revised manuscript, that we can answer. Otherwise, we humbly ask the Reviewer to consider revising their score.

---

> > > ### Comment · Reviewer_yCC2 · 2024-12-02
> > > **Response to Authors**
> > >
> > > Thanks for the responses. Some of my concerns were cleared. However, I still think some solid theoretical analysis is absent, and I would like to keep my original score.

---

> ### Author Response · Authors · 2024-12-04
> **Response to Reviewer yCC2**
>
> We thank the Reviewer for responding to our rebuttal. We are happy to hear that we were able to clarify most of the Reviewer's concerns. As for the theoretical grounding of our work, we describe the theoretical motivation for the DeFaith measure below.
>
> The choice of the DeFaith measure formulation is based on the work of Zhang and Sprites and our observations on the properties of NN-based synthetic data. Zhang and Sprites seek a correct stronger definition of Faithfulness to provide guarantees of uniform consistency for causal structure inference. They formulate $\lambda$-Strong-Faithfulness in the following manner:
>
> [Zhang & Sprites]: A distribution $P$ is said to be $\lambda$-Strong-Faithful to a DAG $G$ with observed variables V if for any $a,b \in V$ and $S \subseteq V \setminus \{a, b\}$
>
> $$ a \text{ is d-connected to} \space b \space \text{given} \space S \space \text{in} \space G \iff |\rho(a,b|S)| > \lambda $$
>
> where $\rho(a,b|S)$ denotes conditional correlation between variables $a$, $b$ given set $S$.
> Note that 0-Strong-Faithfulness is just regular Faithfulness.
>
> Further, Zhang and Sprites prove that when $\lambda$-Strong-Faithfulness is assumed causal structure can be inferred with uniform consistency. Additionally, crucially for our work, $\lambda$ can serve as a notion of the scale of Faithfulness violation. They write *“a faithful but close to unfaithful distribution may be said to be unstable in the sense that some dependence relations may be destroyed by a slight change in parameterization. In this sense, the $\lambda$ in the $\lambda$-Strong-Faithfulness serves as a rough index of stability.”* [Zhang & Sprites, Section 3.2]
>
> We first aimed at using $\lambda$-Strong-Faithfulness as a Faithfulness violation measure in our paper. However, we observed that $\lambda$-Strong-Faithful distributions are extremely rare in the non-linear setting. Note that for $\lambda$-Strong-faithfulness to be fulfilled there has to be a threshold $\lambda$ for which the correlation between d-connected nodes is higher than $\lambda$, and the correlation between d-separated nodes is lower than $\lambda$.
> Since we were having a hard time generating distributions that fulfill this property, we decided to use the following definition of DeFaith:
>
> $\textit{DeFaith}(G) = \underset{a, b \in V, S \subseteq V \setminus \{a, b\}}{\mathit{AUROC}} (1 - |\rho(a, b | S)|, a \space \text{ is d-separated from} \space b \space \text{given} \space  S \space \text{in} \space G)$
>
> This formulation scores how close the sampled distribution is to fulfilling $\lambda$-Strong-Faithfulness.
> Note that acquiring a score of 1 in the DeFaith metric means that a distribution is $\lambda$-Strong-Faithful.
>
> We hope that this explanation provides a theoretical link between the DeFaith measure and the Faithfulness assumption that the Reviewer was seeking. We will include it in the camera-ready version of the paper.
>
> [Zhang & Sprites] Jiji Zhang, Peter Spirtes: Strong Faithfulness and Uniform Consistency in Causal Inference. UAI 2003: 632-639

---

### Official Review · Reviewer_CQrQ · 2024-11-05

**Soundness:** 3
**Presentation:** 2
**Contribution:** 2
**Rating:** 5
**Confidence:** 3

**Summary:**

This paper experimentally demonstrates that violating the faithfulness assumption is the key obstacle in improving the performance of existing neural causal discovery methods. In this paper, the authors introduce a metric to measure the degree of faithfulness violation, and also proposed a method to compute an experimental bound on the causal discovery performance.

**Strengths:**

- I totally agree with the authors on the assertion that faithfulness violations might be the key reason why the performance within the current causal discovery paradigm is limited to a large extent. I am happy to see the experimental support presented in this work.

- The proposed metric of faithfulness violations also makes sense

- The paper is well organized in a logical manner, and is easy to follow.

**Weaknesses:**

- The analyses presented in the paper are only experimental, lacking a theoretical foundation. Although they provide some insights, it might be not convincing enough to generalize to other scenarios.

- The proposed method to compute experimental upper bound is highly unlikely to scale up.

**Questions:**

- Why assuming additive noise SCMs in the paper?

- How general is the proposed faithfulness metric? At what conditions does it work?

- Why not choosing NOTEARS or its variants as baselines? I think NOTEARS might be one of the most well-known neural causal discovery methods?

- Since MLPs have universally strong fitting abilities, the loss easily converges to zero or a very small value when training a network for each parent set. In this case, it is possible that the log-likelihood losses of many DAGs lead to zero or a very small value or have very similar value. If so, how to identify the true DAG?

---

> ### Author Response · Authors · 2024-11-21
> **Response to Reviewer CQrQ**
>
> We would like to express our sincere gratitude to the reviewer for their thoughtful and constructive feedback. We appreciate that the reviewer agrees with the assertion that faithfulness violations may be a key limitation within the current causal discovery paradigm. We are also glad that the Reviewer found the proposed metric for assessing faithfulness violations to be a sensible approach, and the paper well-organized and easy to follow.
> Below, we address the reviewers' questions and concerns, providing further clarification and responses.
>
> **Ad W1**: We agree that theoretical analysis is always useful. However, due to a challenging topic, we first chose to provide experimental evidence supporting our hypothesis. We would like to point out that the data we used was generated using neural networks, which can express a general class of functions and therefore cover a wide range of setups. We would be happy to further develop a theoretical explanation in the future.
>
> **Ad W2**: Yes, the NN-opt method is not scalable, however, it allows us to gain some insights into causal discovery that are otherwise impossible. We included a detailed discussion about the purpose of the NN-opt method in the general response.
>
> **Ad Q1**: We decided to use it to be able to compare a wide set of methods. We included a more detailed discussion and justification in the general response.
>
> **Ad Q2**: In our metric, we aggregate the results of independence tests measured using Spearman correlation. It can be computed for any distribution and graph (unlike $\lambda$-faithfulness [1]). Its limitations are derived from properties of the Spearman correlation coefficient.  Assuming that Spearman correlation is never 0 when nodes are correlated, when faithfulness is fulfilled the DeFaith metric will return a value of 1. For further discussion about the DeFaith metric, please refer to the section with an explanation of the DeFaith measure in the general response.
>
> **Ad Q3**: While NOTEARS is indeed one of the most well-known neural causal discovery, it limits the functional model, however, we tested the DCDI method, which is its successor NOTEARS. For details, please refer to the section regarding methods selection in the general response.
>
> **Ad Q4**: To address this concern, we incorporated an ensemble of neural networks to ensure more robust scoring. Additionally, we analyzed the number of graphs with better scores to evaluate the extent of the problem. This analysis demonstrated that there are many graphs with higher scores than the ground true graph.
>
>
> If any questions remain unanswered or if there are further concerns, we would be happy to provide additional clarification. Otherwise, we would kindly ask the reviewer to consider revising the score in light of the responses provided. Thank you again for your valuable feedback and thoughtful review.
>
> References:
>
> [1] Jiji Zhang, Peter Spirtes: Strong Faithfulness and Uniform Consistency in Causal Inference. UAI 2003: 632-639

---

> > ### Author Response · Authors · 2024-11-30
> >
> > We sincerely appreciate the time and effort the Reviewer has dedicated to reviewing our manuscript. In response to the Reviewer’s questions and concerns, we have provided detailed answers and made corresponding improvements to the manuscript based on their feedback. Please let us know if there are any further comments or concerns regarding our rebuttal or the revised manuscript, that we can answer. Otherwise, we humbly ask the Reviewer to consider revising their score.

---

### Official Review · Reviewer_YhZG · 2024-11-07

**Soundness:** 3
**Presentation:** 3
**Contribution:** 2
**Rating:** 6
**Confidence:** 3

**Summary:**

This paper posits that the primary obstacle in neural causal discovery is the faithfulness assumption frequently used by modern neural approaches. It examines the impact of violations of this assumption both qualitatively and quantitatively, and provides a unified evaluation framework to facilitate further research.

**Strengths:**

- The main claim of this work is that progress in causal discovery requires moving beyond the faithfulness assumption, which is a reasonable standpoint. The key experimental findings are:
 >- despite advancements in causal discovery over the past few years, $ESHD_{CPDAG}$ and $F1-Score_{CPDAG}$ metrics do not improve significantly.
>-  structure discovery accuracy does not scale with the amount of data.
>-  variations in MLP architecture have minimal impact on performance.

-  develop techniques to measure how faithfulness violations degrade performance (Figure 4 is particularly interesting) and set an
upper bound for current benchmarks.

-  The paper is well-written.

**Weaknesses:**

- Line 71 : $U_{i}$ is assumed to be 1-dimensional.  Justify.
- Please justify the additive noise assumption.
- > Faithfulness assumption can be violated, for example, in a situation when paths cancel each other effects out, leading to statistical independence despite an existing causal relationship.

**Give concrete example**
 - Expected SHD between CPDAGs has been discussed; there is no discussion about $F1-Score_{CPDAG}$
 -  Why do Table 1 and Section 3.2  miss the result on ER(5, 1)?
 - why do you exclude *DIFFUSION MODELS FOR CAUSAL DISCOVERY
VIA TOPOLOGICAL ORDERING* in your analysis?

- > To provide a comprehensive evaluation, we explored architectures with 1, 2, and 3 layers, configured
with 4, 8, and 16 hidden units.

**why do you think this is sufficient to conclude *variations in MLP architecture have minimal impact on performance* ?**

- The faithfulness metric, denoted DeFaith needs more detailed discussion. For example,
>  To address this, we introduce a degree of faithfulness metric, denoted DeFaith, to measure how well statistical dependencies correspond to the true graph’s d-separation properties. Inspired by Zhang & Spirtes (2003), we use Spearman’s rank correlation coefficient to quantify the conditional dependencies in the dataset. We define a predictor of d-separation based on the coefficient.

**Please clearly state what is the predictor here.**
**More discussion is needed around the quality of this predictor itself**.

- The paper does not address how to design neural causal discovery methods without the faithfulness assumption.

**Questions:**

Please see Weaknesses

---

> ### Author Response · Authors · 2024-11-21
> **Response to Reviewer YhZG**
>
> We sincerely thank the reviewer for their thoughtful review and for taking the time to evaluate our work. We are grateful that you recognized the strengths of our paper, including the importance of moving away from the faithfulness assumption, the valuable insights into how limited faithfulness impacts performance, and the clarity of the writing.
> Below, we provide detailed responses to the reviewer's questions and address the concerns raised.
>
>
> **Ad W1**: We say that $V$ is a set of endogenous variables and each $V_i$ has an exogenous noise variable $U_i$. To the best of our knowledge this is a standard definition of an SCM. If anything remains unclear we kindly ask the reviewer to rephrase the question so that we can provide an explanation.
>
> **Ad W2**: We decided to use it to be able to compare a wide set of methods. We included a more detailed discussion and justification in the general response.
>
> **Ad W3**: We added a concrete example of a simple graph, where faithfulness can be easily violated to Appendix A2 and mention it in Section 2.
>
> **Ad W4**: We added a short explanation of the F1-score metric to the main text and a more detailed description of our evaluation methodology and justification for the formulation of metrics in appendix D.
>
> **Ad W5**: We added missing results to the Table 1.
>
> **Ad W6**: DiffAN is indeed an interesting method, offering flexibility by not constraining the neural network architecture and scaling well with graph dimensionality. However, the pruning step relies on sparse regression and to ensure the global score optimum lies in the true MEC class, we would have to pose additional constraints, which is out of scope of this work.
>
> **Ad W7**: We are grateful for the suggestion to further explore variations in the MLP architecture. To address this, we expanded our testing by adding more architectural variations. Inspired by BayesDAG, we also implemented layer normalization and residual connections to assess their impact. We conducted additional experiments on both the best-performing model ([4, 4]) and the largest model ([8, 8, 8]). Please note that in all our experiments, the size of networks was similar to the ones proposed in the literature: in DCDI it was [16, 16], for SDCD it was [10, 10], for DiBS [5, 5] and for BayesDAG it was a two layer network with a hidden size varying with dimensionality. The effects of the comparison are discussed in section 3.3 and the plot was added to Appendix C.3 of the revised paper.  We show, there is no significant and consistent improvement across all networks, supporting our initial conclusion that variations in MLP architecture have minimal impact on performance.
>
> **Ad W8**: We define a predictor that classifies nodes as independent if conditional Spearman’s rank correlation coefficient computed based on a dataset D is smaller than a certain threshold. We improved the measure description in the revised version of the paper and provided additional explanations in the general response. If anything remains obscure please let us know.
>
> **Ad W9**: Indeed, designing neural causal discovery methods without relying on the faithfulness assumption is a challenging and interesting area of research. While we emphasize its importance, addressing this topic in detail is beyond the scope of the current paper. We consider it a promising direction for future work, and describe existing approaches in the related work section.
>
> If any questions remain unanswered or if there are further concerns, we would be happy to provide additional clarification. Otherwise, we would kindly ask the reviewer to consider revising the score in light of the responses provided. Thank you again for your valuable feedback and thoughtful review.

---

> ### Comment · Reviewer_YhZG · 2024-11-29
>
> I would like to thank the authors for their response. I am revising my rating.

---

### Author Response · Authors · 2024-11-21
**General response to the Reviewers**

We would like to thank all Reviewers for taking the time to review our work and providing us with insightful feedback. We are glad that the Reviewers appreciated the strengths of our paper, the importance of the subject (Reviewer yCC2, CV1a), the straightforward and interesting formulation of the research problem (Reviewers YhZG, CQrQ, yCC2, CV1a), the technique to measure faithfulness violations (Reviewers YhZG, CQrQ). We are glad to learn that the paper is well-written (Reviewers YhZG, CQrQ, CV1a), easy to follow (Reviewer CQrQ), and that the work is well presented (Reviewer yCC2).
We have carefully considered all feedback provided by the Reviewers and prepared a new revision of the paper based on it. New changes are marked with the red color.
We have identified a set of common concerns that we would like to address jointly:

**Justification for the selection of methods**

Reviewers CQrQ and  yCC2 raised concerns about the suite of methods (DCDI, SDCD, DiBS and BayesDAG) used in our work. We claim that the selected suit adequately covers also NO-TEARS, NO-BEARS, NO-CURL, GRAN-DAG, SCORE, DAGMA, and DCDFG. Below, we provide a detailed explanation, which is now included in Appendix E and hinted at in Section 2.

NO-TEARS is the first method to use augmented Lagrangian and differentiable constraints to enforce DAGness. However, suggested formulation entangles functional and structural parameters, making NO-TEARS applicable only with linear models or restricted neural networks. The NO-TEARS method was improved in GRAN-DAG (introduces separate adjacency matrix and sampling based on gumbel softmax) and then in DCDI (accounts for interventional data). We chose to use DCDI as it the most developed method in this line of work and has clean implementation.

An interesting line of work shows articles introducing methods such as NO-BEARS and DAGMA, that were focused on improving the acyclicity constraint introduced in NO-TEARS, all proposed constraints were unified in the SDCD paper, and a new constraint was proposed, that was shown to perform the best. Additionally, SDCD is also compared against SCORE and DCDFG again presenting better performance.

The two other methods are from the class of Bayesian approaches. DiBS method is selected as a Bayesian approach that uses classic NO-TEARS-based regularization embedded in its prior. The BayesDAG is based on the NO-CURL parametrization of DAGs and provides improvements to the optimization pipeline (uses MCMC instead of SVGD).

We argue that this selection of four methods summarizes various research directions and improvements explored in neural causal discovery over the last four years and well represents the spectrum of existing approaches. If the Reviewers think this comparison is too narrow, we kindly ask the Reviewer to specifically name additional methods that they think it is essential we compare against.


**Clarifications regarding NN-opt efficiency and applicability**

To clarify, NN-opt is a brute force technique intended to be as powerful as possible in causal discovery. The price to pay is efficiency. NN-opt is helpful as an upper-bound benchmark but is not practical to use. To avoid confusion, we put additional information about this in Section 5.

To develop NN-opt we identify two key elements (1) modeling functional relationships and (2) search over the dag space. In practical methods, these are intertwined and solved in an approximate fashion. NN–opt method disentangles those two objectives trading additional compute for more accurate results.
1. The modeling of functional relationships was done by training an ensemble model for each parent set. Such an approach provides a more robust approximation of the score.
2. The exhaustive search removes challenges related to the optimization scheme

---

> ### Author Response · Authors · 2024-11-21
>
> **Explanation of the DeFaith measure**
>
> We would like to thank the Reviewers YhZG and CQrC for appreciating the effectiveness of the DeFaith metric in assessing the influence of the faithfulness violation on the performance. Below, we provide a more detailed description of the metric together with examples and intuition to answer the questions raised by the Reviewers YCC2 and CV1a. We also adjusted the measure description in Section 4 of the revised paper accordingly.
>
> The faithfulness assumption translates into the set of conditional independence statements that need to be satisfied. The proposed DeFaith metric aggregates results of those independence tests. It can be viewed as consisting of two parts: (1)measuring degree of dependence, which is done by calculating conditional Spearman correlation and (2) aggregating the results of all tests, which is done using AUROC.
>
> The spearman correlation coefficient is commonly used to quantify dependence in the non linear data. We selected this method due to its popularity and well known limitations. We would like to point out that we do not strive for novelty in our definition of the DeFaith metric. We try to resort to common and well-tested methods. Our original contribution lies in showing a strong correlation between the DeFaith measure with the average performance of neural causal discovery methods.
>
> Original faithfulness formulation requires all independence tests to align with d-separation statements. However, we never observe this property empirically, thus we use the AUROC to quantify the faithfulness violations on a continuous scale.
>
> Note, that if we assume that Spearman correlation is never 0 when nodes are correlated, then when faithfulness is fulfilled the DeFaith metric will return a value of 1. When the tests do not agree with d-separation statements from the graph better than random guessing then AUROC = 0.5.
>
> **Clarification for additive noise assumption**
>
> We adopt the additive noise assumption as it is explicitly used by many causal discovery methods, including BayesDAG and DiBS, which do not work in different regimes, ensuring fair comparison across approaches. DCDI and SDCD can handle non-additive noise, employing normalizing flows for this purpose, but such methods are more complex and less common, limiting comparability.
>
> We hope that the above points are useful in reducing the confusion, but if anything remains obscure, we kindly ask the Reviewers to point it out to us. For specific answers to the Reviewers' individual concerns, please see our individual answers posted as separate comments.

---

### Author Response · Authors · 2024-11-28
**Manuscript Revision and Follow-Up on Reviewer's Comments**

We sincerely thank Reviewer CV1a for their valuable feedback and for engaging in the discussion. Based on their suggestions, we have made further improvements to the manuscript.

In the revised version, we have included an additional experiment on real-world structures to further substantiate our findings. Additionally, we extended Figure 2 with results on 25K samples. We recomputed all results for DiBS on graphs ER(10,2) as we realized that the set of hyperparameters used previously was not optimal as one of them was incorrectly introduced into the configuration file.

Despite these updates, the main conclusion from section 3 remains unchanged. The method's performance remains unsatisfactory and there is no consistent pattern of improvement in the ESHD_CPDAG metric as observational sample size increases — this holds true for both synthetic and real-world structures. Section 3 has been updated accordingly to reflect the findings from the new experiments.

We kindly invite Reviewers YhZG, CQrQ, and yCC2 to share their responses to our rebuttal. We are happy to provide additional clarifications or further improvements to the paper if needed. Alternatively, if the reviewers are satisfied with our responses and the changes made so far, we respectfully request that they consider revising their scores.

---

### Author Response · Authors · 2024-12-04
**Final Thank You and Rebuttal Summary**

Once again, we would like to express our gratitude for the time and effort the Reviewers took to review our work and for the useful feedback provided. With the Reviewers' help we have been able to provide numerous improvements to the manuscript:
* We included additional experiments to support results from section 3, as suggested by Reviewers YhZG and CV1a. We provided results for benchmarking in higher sample sizes, more network architecture ablations, and additional results on real-world structures. All new results confirm our observations.
* We improved the descriptions of the DeFaith measure (Section 4), NN-opt method (Section 5), and metrics explanation (Section 2)  based on the provided comments.
* By the suggestion of the Reviewers YhZG, CQrQ, and yCC2, we provided additional justification for the selection of methods and metrics and an example of faithfulness violation in Appendices A, D, and E.

We thank the Reviewers YhZG and CV1a for acknowledging that applied changes significantly improved the overall quality of the paper by increasing their scores!

The Reviewer yCC2 signified the need for theoretical grounding of the DeFaith measure. We have just provided a detailed response explaining the theoretical foundation of our approach. We hope the answer is exhaustive, and we will include a similar explanation in the camera-ready version.

We have carefully considered all comments of the Reviewer CQrQ and provided detailed responses for all of them. We hope we have addressed all their concerns, though we regret that we weren't given the opportunity to correct any ambiguities in a discussion.

Thank you for your thoughtful comments and suggestions, which have significantly enhanced the quality of our paper. We hope that the improved manuscript meets the expectations of the Reviewers and the broader research community.

---

### Meta-Review · Area_Chair_C9yG · 2024-12-20

**Metareview:**

The paper investigates the impact of faithfulness violation on current causal learning methods. The motivations for this assumption are clear; what happens when it fails (and to a lesser extent, why) is explored in the paper. The study focuses on neural causal learning methods.

Two lessons learned are: i) the data amount does help to some extent (then the performance reaches a plateau); ii) the complexity of the neural architecture does not matter much.
One interesting contribution is a new indicator, measuring the faithfulness violation, called DeFaith. This indicator is shown to be correlated to the performance (causal graph identification). Note however that this indicator requires the ground truth to be known.

**Additional Comments On Reviewer Discussion:**

A first issue raised by reviewers concerns the choice of the neural causal learning baseline methods (now discussed in Appendix E); however, the restriction to neural approaches is puzzling as  it seems that PC should be the method most impacted by faithfulness violations.

The experiments are limited in size and degree.

The area chair believes that the paper addresses a very significant issue; however, the lessons learned are not yet sufficient for publication in a venue like ICLR.

---

### Decision · Program_Chairs · 2025-01-22

Reject